# Studying Parameters Affecting Accumulation of Chilling Units Required for Olive Winter Flower Induction

**DOI:** 10.3390/plants12081714

**Published:** 2023-04-20

**Authors:** Chaim Engelen, Tahel Wechsler, Ortal Bakhshian, Ilan Smoly, Idan Flaks, Tamar Friedlander, Giora Ben-Ari, Alon Samach

**Affiliations:** 1The Robert H. Smith Institute of Plant Sciences and Genetics in Agriculture, The Robert H. Smith Faculty of Agriculture, Food, and Environment, The Hebrew University of Jerusalem, P.O. Box 12, Rehovot 7610001, Israel; 2Institute of Plant Sciences, Agricultural Research Organization (ARO), Volcani Center, Rishon LeZion 7528809, Israel

**Keywords:** flowering, *Olea europaea*

## Abstract

With global warming, mean winter temperatures are predicted to increase. Therefore, understanding how warmer winters will affect the levels of olive flower induction is essential for predicting the future sustainability of olive oil production under different climactic scenarios. Here, we studied the effect of fruit load, forced drought in winter, and different winter temperature regimes on olive flower induction using several cultivars. We show the necessity of studying trees with no previous fruit load as well as provide evidence that soil water content during winter does not significantly affect the expression of an FT-encoding gene in leaves and the subsequent rate of flower induction. We collected yearly flowering data for 5 cultivars for 9 to 11 winters, altogether 48 data sets. Analyzing hourly temperatures from these winters, we made initial attempts to provide an efficient method to calculate accumulated chill units that are then correlated with the level of flower induction in olives. While the new models tested here appear to predict the positive contribution of cold temperatures, they lack in accurately predicting the reduction in cold units caused by warm temperatures occurring during winter.

## 1. Introduction

The olive (*Olea europaea* ssp. europaea var. europaea) is a perennial evergreen tree originally domesticated in the Mediterranean Basin for its fruits [1], mostly used for table olives and olive oil. Olive oil is the main source of dietary fat in the Mediterranean diet. This diet has several health benefits [2], including reducing major cardiovascular events [3]. The current global annual consumption of olive oil is ~3.2 million tons (International Olive Council 2021), with close to 95 percent of production in regions surrounding the Mediterranean Sea.

The average annual olive fruit yield for a whole region can vary between 5–30 tons of olives per hectare [4]. The amount of olive oil, extracted from the fruit pericarp, will be much less. In one example, the percent of olive oil commercially extracted from fresh olives varied between 8–36%, depending on cultivar and environmental conditions [5]. Traditionally, rainfed olives are planted at a low density of 100 trees per hectare. Recently, irrigated olives are being planted at much higher densities reaching up to 2500 trees per hectare. A mature olive tree, planted at a density of 250 trees per hectare, is estimated to produce up to 0.5 million flowers per year [6]. The average number of flowers on an inflorescence varies among genotypes [7,8,9]. Within a genotype, the number of flowers on an inflorescence can also vary. For example, one study measured in cv. ‘Barnea’ between 13–27 flowers per inflorescence, with most inflorescences carrying 19 flowers [8]. In the above example, a tree with 0.5 million flowers would contain more than 26,000 inflorescences. Inflorescences are formed at the end of winter from lateral meristems on year-old shoots (vegetative growth that began the previous spring) [10]. Olive shoots normally have opposite phyllotaxy, and, as such, the potential number of inflorescences that can develop is twice the number of new nodes formed between spring and fall. In a specific cultivar, at a specific location, the percentage of lateral meristems that will form inflorescences (*i*) can be highly variable between different years [11]. An olive inflorescence normally contains male and perfect (bisexual) flowers. Most bisexual flowers will not develop into fruits. The ratio between the two types of flowers, as well as the average amount of fruit retained on an inflorescence is both genotype and environment dependent [8,9,11]. Olive self-incompatibility [12] is one of the factors contributing to low mean fruit number per inflorescence. 

Thus, the three major developmental factors affecting olive productivity are the number of new buds formed on one-year-old shoots (*n*), the percentage of lateral meristems that will develop an inflorescence (transition to flowering; *i*), and the percentage of inflorescences forming and retaining fruit [11]. Here, we focus on factors that affect *i*. 

An olive tree in fall will have several thousand lateral meristems, and whether or not they will form inflorescences at the end of winter depends on two major factors: the fruit load on the tree at the end of summer and the temperatures the trees will be exposed to during winter, termed here as the ‘Winter Temperature Regime’ (WTR). This is because flower induction in olives during winter is inhibited by the ‘memory’ of fruit load [10] causing an effect termed ‘alternate bearing’ [4]. Olive flower induction during winter requires cold winter temperatures and is not influenced by day length [10,13,14,15]. The term ‘Chilling Requirement’ is often used in the literature, in some cases assuming that once a certain requirement is reached, flowering will occur. Indeed, there are some artificial winter conditions leading to no flowering. For example, potted olive trees from several cultivars subjected to WTRs with a minimum daily temperature of 15.5 °C [8], constant temperatures of 18 °C or 4 °C [13], or day/night temperature cycles of 28/22 °C [10] did not flower. Still, in WTRs that lead to flowering, *I* levels can vary tremendously, suggesting that there is no specific chilling requirement that, once met, causes all lateral meristems to form inflorescences and, if not met, leads to no flowering at all.

What temperatures induce flowering in olive trees? Replacing a natural outdoor Los Angeles (California) winter with 112 days of constant 10 °C or 13 °C temperatures caused slightly lower levels of flowering in Olive cv. ‘Manzanillo’ [13]. Shortening the period of 13 °C cold exposure to 49 days did not induce flowering, and a 95-day exposure appeared to cause similar flowering as longer exposures [13]. Furthermore, 68 days at a constant 12.5 °C or a fluctuating 21/12.5 °C for 4/20 h respectively caused high flowering in ‘Manzanillo’ [16]. Replacing a natural, outdoor Rehovot (Israel) winter with 89 days of day/night temperature cycles of 16/10 °C resulted in much higher *i* values in cv. ‘Barnea’ [10]. Shortening the natural winter period by moving potted trees to a heated glasshouse (with a daily minimum of 15 °C) in mid-December caused a significant reduction in *i* [10]. Artificially exposing olive trees to similar inductive cold temperatures as described above in other seasons can cause out-of-season flowering [10,17]. These experiments, although conducted with different genotypes, as well as different control outdoor winter conditions and tree conditions, point to a mechanism in which a winter period of 13–15 weeks with a temperature range above 5 °C and below 15.5 °C promotes flower induction in olives. In order to quantify this cold accumulation, the idea of hypothetical ‘chilling units’ (CUs) accumulated during the cold season was explored with the assumption that a minimal number of CUs is required to induce flowering. Using parameters obtained by the Hartmann group [18], 12.5 °C was suggested as a ‘compensation point’ around which CUs are accumulated, such that days with a maximum temperature between 12.5 and 21.2 °C and a minimum temperature between 0 and 12.5 °C are considered days that contribute to flower induction [19]. Using a weighted function that sums up the days matching this criterion showed values between 10 and 140 in WTRs of different regions in Texas [19] and between 0 and 200 in different regions of Argentina, Mexico, the USA, and Europe [20]. 

Flower induction in different plant species is largely mediated by the accumulation of the FLOWERING LOCUS T (FT) protein in leaves and its movement to meristems where it triggers the transition to flowering [21]. Species differ in the environmental and internal signals that trigger the accumulation of *FT* transcripts in leaves [21]. In olives, the expression of two genes encoding proteins similar to FT increases in leaves during winter and in controlled conditions of 16/10 °C provided in winter or other seasons [10,11]. In trees with high fruit load before winter, the accumulation of FT-encoding transcripts during winter is suppressed [10]. A similar ‘cold temperature inducing’ and ‘fruit load inhibiting’ effect on *FT* expression was identified in other evergreen fruit trees in which flowering is induced by cold WTRs such as mango [22,23], citrus [24], litchi [25], and avocado [26]. In all the above cases, each of these perennial tree species has thousands of meristems, each with the potential to transition to an inflorescence. While the decision for each meristem is binary, *i* values are quantitative. 

Another developmental switch that requires long exposures to cold temperatures is bud release from dormancy in deciduous perennial trees [27]. Here, several mathematical models try to estimate the number of CUs accumulated during WTRs. The simple “Chilling Hours” model sums up all hours in which temperatures were between 0 and 7.2 °C [28]. In the more advanced “Utah Model” [29,30], different temperatures within that range can have different CU contributions. 

In the past, several olive researchers described olive phenology as similar to deciduous perennial trees such as apples with flower induction occurring in late summer and cold WTRs required for breaking bud dormancy [31,32]. As a result, several published chilling models in olive discuss the process of bud dormancy release and use temperatures and models defined for deciduous trees. One simple model counts the number of hours below a certain base temperature: 7.2 °C [33,34]. Another, termed as the ‘’trigonometric approach’, predicts different base temperatures depending on the chilling requirement of a cultivar: 7.0 °C, 9.5 °C, and 13 °C for high, medium, and low chilling requirements, respectively [35]. The De Melo-Abreu (DMA) model [36] initially developed for predicting the olive flowering date is similar in structure to the Utah model, with each hourly temperature contributing either a positive or negative amount of CU. The optimum temperature for chilling accumulation in this model is 7.3 °C, and temperatures above 15.9 °C nullify CUs, with maximum nullification reached at 20.7 °C and above. The period measured for CU accumulation in the northern hemisphere begins on October 1st and ends once a threshold of CUs was reached, considered the end of ‘endodormancy’. Temperatures later on in spring are translated into Growth Degree Days (GDD) and affect the speed at which anthesis occurs, with warmer temperatures promoting earlier flowering. The model does not calculate the percentage of meristems forming inflorescences. It predicts whether the ‘chilling requirement’ for a certain cultivar was fulfilled by reaching a certain threshold of CUs, leading to ‘normal’ flowering. Researchers stated that ‘Arbequina’ requires 339 CUs, while ‘Frantoio’ and ‘Leccino’ require 612 and 671 respectively [37]. This model has helped predict the success of growing olives in new climatic regions, such as Northwest Argentina in South America [37]. Its parameters have been incorporated in a more expansive model aimed at predicting olive oil production [38,39,40]. Others [41,42] have used a dynamic model in olives that was originally developed for bud dormancy release in peach [43,44], as described below, in which optimal CUs are obtained at temperatures of 6–8 °C. 

Recent publications focus on the effect of climate on the date of olive flowering [36,42]. Within traditional olive growing regions as well as other world regions, warmer WTRs will mean significantly less flowering and yield for some of the more popular olive cultivars [35,38], although some predictions show no potential reduction in flowering levels [40]. With this unknown future for olive production, a more precise model that can reliably predict the level of flowering in olives based on hourly winter temperatures may be a timely resource for both olive growers and researchers.

To construct such a flowering prediction model, we first need to disentangle the relative contribution of different internal and external factors to flowering. In a set of controlled experiments, we quantified the effects of the previous year’s fruit load, drought, and winter temperatures. Since after a year with heavy fruit load, olive trees will not flower even when exposed to a cold winter [10], trees used to calibrate this model may require full or partial fruitlet removal in spring. Here, we measured the return flowering of trees in which fruitlets were removed from one sector of the tree, compared to fruitlet removal from the entire tree. As we noticed that flowering in some cultivars is affected by the memory of fruit load in a nearby sector, we continued our research on trees in which all fruitlets were removed in spring. Water deficit before or during winter promotes flowering in several evergreen woody perennial species that require winter cold for flower induction [45]. These include several citrus species [46], *Shorea* species [47], litchi (*Litchi chinensis*) [48], and star fruit (*Averrhoa carambola*) [49]. The Mediterranean rainy season coincides with cold temperatures during winter, and since most olives are rainfed, it is not clear whether winters with less rain will affect olive flower induction. Here, we tested the effect of forced drought (FD) during winter on the flowering of different olive varieties. Further, we measured *i* values in 5 cultivars exposed to 9–11 different WTRs. Based on hourly temperature collection, we tested different CU models and their ability to predict *i* values. 

## 2. Results

### 2.1. Does the Memory of Fruit Load in a Different Tree Sector Affect Flowering?

An accurate flowering model relies on precise flowering data from trees exposed to different WTRs. Olive flowering is also influenced by previous fruit load [10]. In an experiment performed on potted ‘Barnea’ trees in which artificial winter conditions were provided during fruit development (during summer and, therefore, out of season), fruit-bearing branches flowered less than branches without fruit (Appendix A). In our next experiment, we asked whether it would it be sufficient to remove fruitlets from a major branch in spring (before June 1st) and record *i* values during the following spring only on branches within that particular ‘no fruit sector’. We hypothesized that *i* values in the ‘no fruit sector’ would be similar to those of trees with no fruits at all. In other words, we expected that the effect of fruit load is mostly localized to the fruit-bearing branch, and that cross-branch effects are negligible. We measured *i* in ‘Manzanillo’ and ‘Coratina’ trees in which fruitlets were removed from the whole tree, compared to trees in which fruitlets were removed from a sector that was approximately one-third of the tree (Section 4.1). Fruit load reduces *n* values on new branches [50]. In ‘Manzanillo’ *n* values were similar between trees with no fruits and the ‘no fruit’ sectors (Figure 1A). As expected, on branches with two or four fruits, *n* values were reduced compared to the ‘no fruit’ sector (Figure 1A). In ‘Coratina’ *n* values were higher in trees with no fruits compared to ‘no fruit’ sectors (Figure 1A). Sectors with no fruit had similar *n* values as sectors with two or four fruits. Thus, in ‘Coratina’ in this experiment, we find that fruit load on one sector of the tree could affect *n* in the other sectors that did not have fruit (Figure 1A, Appendix A).

Fruit load in one sector also significantly influenced flowering in other tree sectors and, in particular, in the ‘no-fruit’ sector in ‘Coratina’, reducing *i* from ~80% in ‘no fruit trees’ to ~5% in ‘no fruit’ sectors of trees with fruits in other sectors (Figure 1B). Thus, adjacent sectors carrying fruit load significantly reduced both *n* and *i* in sectors without fruit in ‘Coratina’ (Appendix A). This effect of fruit load in one sector on *i* in the ‘no-fruit’ sectors was not noticed in ‘Manzanillo’ trees (Figure 1B). Hence, to eliminate the fruit load effect and focus on winter temperatures, for the following experiments we removed all fruitlets from the trees tested. 

### 2.2. Forced Drought (FD) during Winter Does Not Affect Olive Flower Induction

The amount of rain during Mediterranean winters differs by region and by year, and in rainfed orchards, drier winters cannot be supplemented by irrigation. As drought can contribute to flower induction in other species, the relevance of this parameter in olive flowering needed to be evaluated. We experimented on five olive cultivars: ‘Barnea’, ‘Coratina’, ‘Koroneiki’, ‘Picholine’, and ‘Souri’ grown outdoors in pots in Bet-Dagan, Israel (31°59′26.1″ N, 34°49′10.2″ E) during the winter of 2020–2021 (Section 4.2). For each cultivar, one group was the control that received both regular irrigation as well as natural precipitation. The FD group did not receive any type of irrigation or rainfall beginning 24 November 2020 and ending 12 January 2021, except to avoid mortality (Section 4.2 and Appendix A). Weekly stem water potential values for each of the trees (Appendix A) and average values for trees of the same cultivar (Figure 2) are presented. The stem water potential of irrigated trees did not change significantly during the winter and was similar in all cultivars (Appendix A, Figure 2), with the lowest values for a single tree reaching −1.6 MPa. The FD treatment reduced water potential to ranges below −3 MPa, reaching averages of −4.9 MPa, with single trees reaching extremely low values such as −6.8MPa. Thus, the treatment had a significant effect on tree stem water potential (Figure 2F). In many cases, the leaves of most of the FD-treated trees curled, and some abscised, indicating the stress level of these trees. Interestingly, aside from one tree, FD-treated trees of the ‘Picholine’ cultivar did not reach a water potential below −2.3 MPa (Appendix A), and leaves of these trees appeared to be in a much healthier physiological state compared to other cultivars undergoing the same treatment, suggesting this cultivar was less sensitive to FD conditions. Thus, there was a significant difference between cultivars regarding the effect of FD treatment on water potential (Figure 2F, Appendix A). 

We wanted to verify that the samples collected from the FD-treated trees show changes in expression of genes previously shown to be affected by water stress. We followed the expression of the water-channel-encoding *OePIP2.1* olive gene [51] in leaves at three time points during winter for the different cultivars with or without FD treatment. Indeed, on many occasions, expression of *OePIP2.1* was reduced by the prolonged FD treatment, significantly in ‘Barnea’ (Appendix A). We then studied *OeFT2* gene expression in leaves at three time points during the winter for all five cultivars with or without FD treatment. The expression of this gene in leaves of the ‘Barnea’ cultivar was shown to increase during winter, and its increased expression is associated with high levels of flower induction [10]. In this experiment, expression of *OeFT2* in leaves did indeed increase during winter in ‘Barnea’ trees as well as in all the other cultivars (Figure 3A). *OeFT2* relative level of expression was much higher in ‘Barnea’ compared to ‘Picholine’, ‘Souri’, and ‘Coratina’ (Figure 3A). Interestingly, no significant difference was found in the relative expression of the *OeFT2* gene between the control and FD-treated plants for any of the cultivars on any specific sampling date (Figure 3A, Appendix A).

*i* values were also not affected by the FD treatment (Figure 3B, Appendix A). We note, however, that the winter of 2020–2021, when this experiment was conducted, was relatively warm (see below) such that flowering levels of trees were low to begin with. From the forced-drought experiment, we learn that the FD treatment we conducted had no measurable effect on *i*, and hence does not need to be considered when studying the relationship between CUs and *i* levels under different WTRs. Differences in *i* levels between cultivars were quite pronounced with low flowering in ‘Picholine’ and the highest flowering in ‘Coratina’ and ‘Koroneiki’ (Appendix A). The differences we observed in *OeFT2* levels between cultivars did not correlate with *i* levels (Figure 3A,B). For example, *i* values in ‘Coratina’ were higher compared to ‘Barnea’, yet *OeFT2* levels in ‘Barnea’ were higher compared to ‘Coratina’ (Appendix A).

### 2.3. Different WTRs, CU Calculations, and i Values in Five Cultivars

To better quantify the winter chilling requirements of different cultivars, we collected *i* values for five cultivars (‘Askal’, ‘Barnea’, ‘Coratina’, ‘Manzanillo’, and ‘Nabali’) exposed to between 9 and 11 different natural winters. ‘Barnea’ trees were subjected to an additional treatment beginning with a natural winter, after which the trees were moved to a heated glasshouse in mid-winter (24 December). The number of biological repeats (trees) for a specific combination of year/location/cultivar varied between three and six. Data were collected from 2014 to 2022 in four locations within Israel (Section 4.3). All trees measured had no previous fruit load since all fruitlets that formed on these trees were manually removed before the end of Spring (by 1 June) of every year, as a result of the fruit load experiment reported above. We also collected hourly temperatures during winter in the relevant locations: Gilat, Rehovot, Hanania, Matityahu, and the heated glasshouse (see Section 4.3 for exact locations, as well as temperature data collection). Altogether, we recorded *i* values for 48 treatments and hourly temperatures for 12 experiments (Appendix A). For each location, hourly temperatures were collected from 15 October until 15 March. For the purpose of calculating CUs, the end of winter was determined as either 15 February for warmer locations or March 6th for colder locations (see Section 4.3 for rationale used). 

The DMA model sets maximal CU accumulation at 7.3 °C and suggests that hourly temperatures above 16 °C result in a negative CU contribution, i.e., the negation of accumulated cold units [36] (Figure 4A). The earliest dates on which hourly temperatures below 16 °C were detected varied between winters and locations. Hence this model allows for negative CU accumulation. For example, during the winter of 2017–2018 in Rehovot, the DMA model exhibits negative CU accumulation until 24 December (Figure 4B). To handle such cases, we considered the difference between the CUs accumulated on the last date of measurement and the accumulated units at their minimal level during a specific winter as the net CU accumulation (Figure 4B). Higher values of DMA CUs were accumulated in northern regions (Hanania and Matityahu) compared to southern regions (Rehovot and Gilat) of Israel (Figure 5, Figure 6 and Figure 7). Within Rehovot, accumulated CU levels in different winters ranged between 197 and 493 (Figure 5, Figure 6 and Figure 7). In some cases, we found a poor correlation between DMA CUs and recorded flowering values *i*. For example, while calculated DMA CU levels were higher for the 2019–2020 Rehovot winter (447 CUs) compared to the 2016–2017 Rehovot winter (284 CUs), we find the opposite to be true for *i* levels, which were higher after the 2016–2017 winter compared to the 2019–2020 winter in all cultivars (Figure 5, Figure 6 and Figure 7). This model appears to be less useful in distinguishing between low to medium flowering levels. 

To improve the quality of the model prediction, we defined a new cold accumulation model for which maximum accumulation is shifted to 11 °C instead of 7.3 °C used in the DMA model. This change was based on temperature experiments described in the introduction. Initially, we tried out a simple positive CU model, termed here the ‘Gaussian’ model, in which the degree of cold unit accumulation changes by a gaussian curve (Figure 4A): $k_{0}\left(T\right)=e^{-\frac{{\left(T-\mu \right)}^{2}}{2\times \sigma^{2}}},\;\mathrm{where}:\mu =11\;\left[{}^{\circ}C\right]\;\mathrm{and}\;\sigma =2.65\;\left[1/{}^{\circ}C\right].$ Since this model does not have negative units, CU accumulation during winter is only positive (Figure 4B). Under different WTRs, this model, similar to the DMA model, exhibits higher CUs accumulated in northern regions (Hanania and Matityahu) compared to southern regions (Rehovot and Gilat) of Israel (Figure 5, Figure 6 and Figure 7). 

Within Rehovot, Gaussian CUs accumulated in different winters ranged between 743 to 1071 CUs (Figure 5, Figure 6 and Figure 7). Similar ‘Gaussian’ CUs (1010 CUs) were calculated for the Rehovot 2019–2020 winter as well as the Gilat 2016–2017 winter, though flowering in all cultivars was higher in the Gilat 2016–2017 winter (Figure 5, Figure 6 and Figure 7). A model which ignores the negative effects of warm temperatures during winter is unlikely to correctly predict olive flowering. Hence, as an initial solution, we added a similar negative effect to the ‘Gaussian’ model as shown in the DMA model for temperatures above 16 °C and termed it as the ‘Gaussian + Neg’ model (Figure 4A). We previously found that an artificial temperature regime of 16/10 °C for 53 days from the start of treatment (25 November) until 17 January, two weeks before inflorescence emergence (31 January), resulted 86% flowering in ‘Barnea’ trees [10]. Calculated CUs for this treatment are 581 DMA CUs, 975 ‘Gaussian’ CUs, and 955 ‘Gaussian + Neg’ CUs.

During the 2017–2018 Rehovot winter, ‘Gaussian + Neg’ CU levels reached a minimum on 24 December (Figure 4B). This is not typical of the area and therefore stands out among the data that we collected. To sum up the CUs for a winter of this type, we calculated the difference between the number of units accumulated on the last day of measurement and the accumulated units at their minimum level (24 December) during this winter, a similar method as that proposed in the DMA model (Figure 4B). Since the winters in Matityahu had relatively few hours above 16 °C, the difference in CUs between the ‘Gaussian’ and the ‘Gaussian + Neg’ models was as low as 87–108 units. During the Rehovot 2020–2021 winter, the difference in CUs between the ‘Gaussian’ and the ‘Gaussian + Neg’ models was 393 units. This was due to spells of warm temperatures during this particular winter, leading to the lowest levels of flowering in ‘Coratina’, ‘Manzanillo’, and ‘Nabali’ (Figure 6 and Figure 7, ‘Barnea’ was not recorded during this year). When comparing the DMA and ‘Gaussian +Neg’ CUs in different winters, similar values were reached in the short Rehovot 2017–2018 winter (0 CUs), as well as after the Hanania 2016–2017 winter (1095–1096 CUs). In all other winters, except for the Matityahu 2019–2020 winter, CU calculations using the ‘Gaussian + Neg’ model gave a higher CU value compared to DMA CUs. For hours with temperatures below 9.2 °C, the ‘DMA’ model predicts higher CUs, while for hours above 9.2 °C and below 20.7 °C, the ‘Gaussian + Neg’ model predicts higher CUs. The Matityahu winter of 2019–2020 had the highest number of hours below 9.2 °C, yet *i* levels were not the highest under this WTR for any of the cultivars (Figure 5, Figure 6 and Figure 7). When comparing *i* levels between the two locations during the 2019–2020 winter, *i* levels were significantly higher in Matityahu compared to Rehovot in ‘Barnea’, ‘Coratina’, and ‘Nabali’ (Table 1).

Although the dynamic model uses parameters appropriate for dormancy break in peach [43,44], it has been recently used in estimating olive flowering date [42]. Thus, we also calculated Chilling Portions based on the dynamic model (Section 4.8) for the different winters (Table 2). For the DMA, ‘Gaussian + Neg’, and Dynamic models, Rehovot winters accumulated the fewest CUs compared to other locations (Figure 5, Figure 6 and Figure 7, Table 1 and Table 2). Under Rehovot winters, the cultivar with the highest *i* values was ‘Askal’, and in many Rehovot winters, the cultivar with the lowest *i* values was ‘Barnea’. Average *i* values for ‘Askal’ in different winters ranged between 25 and 97% (Figure 5A, Table 2). Average *i* values for ‘Barnea’ in different winters ranged between 10 and 87% (Figure 5B, Table 2). Significantly higher *i* values in ‘Askal’ compared to ‘Barnea’ were identified in several winters (Table 2). This suggests that the number of CUs required for ‘Askal’ flowering is lower compared to ‘Barnea’. 

## 3. Discussion

The future of commercially viable olive cultivation in the Mediterranean region may be in jeopardy due to global warming [15,40,52]. Here, we focused our study on one biological process in which temperatures affect production: flower induction during winter. Flowering in a certain species can be triggered by different environmental signals such as changes in photoperiod, temperature, or soil moisture. In several evergreen trees, including olives, flowering is induced by cold winter temperatures [10,22,23,24,25,26,47,53]. FT is a universal florigen in plants [54,55,56,57]. Environmental signals that induce flowering in a certain genotype/species normally lead to the accumulation of FT-encoding transcripts in leaves. The FT protein can travel via the phloem to a nearby meristem and change its fate from vegetative to reproductive [55,58]. In olives, cold temperatures during winter lead to the accumulation of transcripts in leaves-encoding FT proteins, as shown here and previously [10,11]. Abnormally warm natural (shown here) or artificial [10,13] winters inhibit flower induction in olives. For each meristem, the decision whether to continue forming leaf primordia or to switch to forming an inflorescence is either a yes or a no answer and rarely a mixed decision (leafy inflorescences). In annuals, once the decision is made, the fate of the whole plant changes. In woody perennials, there can be thousands of meristems, with each one either going through the reproductive transition or remaining vegetative. Consequently, as seen in many of our results, *i* values vary between olive trees of the same genotype exposed to identical winter conditions at the same location. In colder winters, such as the Hanania winter of 2016–2017, *i* values of trees of the same genotype are high and more uniform. Warmer winters may not always cause less flowering in olives. In an experiment performed at an experimental farm of Campus de Rabanales, University of Córdoba, Spain (37°55′ N, 4°43′ W), warming trees by 4 °C starting in January did not cause a significant reduction in flowering intensity [59]. This is likely because increasing temperatures by 4 °C in this region during January–February still allows for sufficient CU accumulation. 

Unfortunately, numerous current olive studies still describe olive winter chilling as a requirement for release from ‘floral bud dormancy’. Using this term, instead of ‘flower induction’, is not only semantic. With the release from dormancy, there is a so-called ‘threshold’ of CUs that needs to be reached for each cultivar, so every winter provides a binary answer regarding whether the threshold was reached or not and then predicts the expected flowering time. While flower induction is also a binary decision for each meristem, *i* values for a tree are a quantitative trait without a threshold. Olive flowering levels are known to vary significantly between years, whereas the change in flowering time may gradually change with climate change. Thus, threshold models are insufficient in providing quantitative answers regarding flowering levels.

It is interesting that, although *i* values vary, the period of anthesis is more or less confined to a certain window of time within a location. This is important for successful fertilization since olives are self-incompatible and wind-pollinated. Since the olive is wind pollinated, the period of anthesis can be determined based on pollen collection using pollen traps [34]. Still, out-of-season flowering in olives can be achieved by artificially exposing plants to cold temperatures in different seasons [10,17]. 

Mature olive fruits are normally harvested in the northern hemisphere during August–December [60,61], with most harvests being before December. Thus, there is normally no fruit on trees during flower induction. The biennial bearing of olives is well known [62,63,64], and its molecular mechanism appears to be the ability of a signal formed by past fruit load to repress the increase in FT-encoding transcripts in leaves during winter [10]. As a result, trees with high fruit load in one year will have very low *i* values the following year [11] and vice versa. These fruit-load-dependent low *i* values will occur even after a cold winter, in which trees that carried little or no fruit reach high *i* values [11]. We show here that even having just two fruits per branch had a significant, inhibitory effect on *i* values in ‘Manzanillo’. Thus, to clarify the effect of different winter temperature regimes (WTRs) on *i*, it is critical to study trees that carried no fruit during the previous year. This was achieved by hand-removing all fruitlets before 1 June. This is quite a labor-intensive task and it leaves the experimental trees with no yield. We speculated that removing all fruitlets from one sector (a main branch which was ~33% of the tree volume) would be sufficient to neutralize the fruit load effect on flowering in that sector, and fruit load in other sectors would not have any lateral effect. Interestingly, this was indeed the case in ‘Manzanillo’ trees, yet in ‘Coratina’ trees, the sector cleaned of fruitlets before June 1st had very low *i* levels compared to nearby trees of the same variety in which all fruitlets were removed. Thus, we concluded that complete fruitlet removal is likely necessary to ensure the correct interpretation of the effect of different WTRs on *i* in at least some of the cultivars. This might not be necessary for much bigger trees, in which sectors might be sufficiently separated from each other so that an inhibitory signal caused by fruit load does not reach the ‘no-fruit’ sector. This hypothesis could be tested in large ‘Coratina’ trees. 

In some evergreen perennial woody trees, flowering is induced by fall or winter drought [46,47,48,49]. Most olive orchards in the Mediterranean region are rainfed. The Mediterranean rainy season coincides with winter olive flower induction with *i* values in rainfed orchards being similar to irrigated orchards of the same genotype [11]. Would drier winters affect olive *i* levels? Various studies have shown that mild drought during winter had no positive or negative effect on flowering [65,66] as long as the drought was not severe in duration or water potential value [65,67,68]. It is difficult to know whether trees growing under rainfed conditions cross this water potential threshold during the inductive period, and, as such, drought could not be completely ruled out as not having an effect on spring olive flowering. Other research shows that there is a possibility that drought can have a positive effect on the flowering percentage of olive trees otherwise not expected to flower or to flower minimally as a result of a lack of sufficient cold winter temperatures or the negative effect of warm winter temperatures. This was shown in an experiment carried out in Peru and Spain [69]. In Peru, plants were expected to have a lower flowering percentage as a result of the local winter temperature profile when compared to those in Spain, which received adequate cold during the winter. The results of this study were that the longer that plants of the ‘Frantoio’ and ‘Criolla’ cultivars were exposed to drought, the higher their flowering percentages were. No difference was found in bud dimensions between buds taken from the experimental site in Peru compared to ones taken from the control site in Cordoba, Spain. The authors suggest that the drought treatment was able to make up for the lesser cold found at the Peru experimental site in comparison to the control site in Spain [69]. We tested this hypothesis here by restricting trees’ access to water for a month, starting 24 November. After a month of receiving no water via irrigation or precipitation, trees were irrigated for three days, and then, plants were not watered for the following 16 days. In most cultivars, this drought treatment significantly reduced water potential, yet strikingly, it did not affect *OeFT2* gene expression in leaves nor *i* values. Thus, it appears that drought conditions during winter do not have a direct effect on *i*. It should be noted that trees under all-year rainfed treatment have reduced *n* levels so that the number of inflorescences they form is significantly lower than irrigated trees [11]. Most research on the effect of drought on productivity is focused on stages after flower induction—for example, during fruit development [59,70,71].

Based on the evidence described in the introduction, we suggest here that the optimal temperature for flower induction in olives is 11 °C. In addition, we suggest that the inductive temperatures for flowering in olives are much better represented by a ‘Gaussian’ model compared to the previously used DMA model. Our results, together with the results of the Hartmann group, suggest that temperatures below 6 °C, considered inductive in the DMA model, are not flower inductive for the olive cultivars tested. We have previously shown that a temperature regime of 16 °C during the day and 10 °C during the night for 89 days is very inductive for olive flowering [10]. While the ’Gaussian’ model might be more accurate than the DMA model in measuring the positive accumulation of CUs, by itself it is not accurate in predicting *i* levels. For example, when comparing ‘Barnea’ *i* levels after the 2016–2017 winter in Gilat or after the 2019–2020 winter in Rehovot, ‘Gaussian’ CU prediction for both winters is 1011 (Table 2), yet *i* levels are ~6 fold higher in Gilat. This is because the ‘Gaussian’ model contains no component considering the negative effect of warm temperatures on CU accumulation. The ‘Gaussian +Neg’ model tested here added a negative accumulation under temperatures above 17 °C, quite similar to the DMA model (Figure 4). The ‘Gaussian + Neg’ model indeed predicted less (51) CUs in the 2019–2020 winter in Rehovot compared to the 2016–2017 winter in Gilat (Table 2), yet this relatively small reduction in CUs does not explain the much lower *i* levels.

One problem with the above models is that hourly temperature measurements for a long winter period are collected and summed up without context. For example, in a winter containing 60 h of 24 °C, the distribution of these warm hours within the winter (clustered, or spread out throughout the whole winter) is not considered in any of these models, even though the distribution of these warm hours likely does affect developmental processes. In the “dynamic model” [43,44] developed for dormancy release, the timing of warm temperature periods within the winter is taken into consideration. The parameters it contains are for peach dormancy break and not for olive flower induction; thus, the values it calvelop such a model, suited for olive flower induction, to establish how warm temperatures within the winters have or do not have an effect on CU accumulation. A model that can only predict the *i* values of trees with no previous fruit load has limitations for practical application. A further step in developing an accurate flowering model is incorporating the effect of previous fruit load on flowering. As shown here, the parameters for the effect of fruit load would likely differ between cultivars.

We observed here several significant differences between cultivars. The ‘Picholine’ cultivar was least affected by the FD treatment compared to the other cultivars, based on the measured water potential values during the FD treatment and by the visual condition of leaves under the FD treatment. On the other hand, more trees from the ‘Coratina’ cultivar required supplemental irrigation to avoid irreparable damage during the FD treatment, compared to the other cultivars. Differences in olive cultivar response to water stress have been reported previously. ‘Koroneiki’ reached lower water potentials compared to ‘Meski’ after 40 days without water [72]. ‘Chetoui’ appears more adapted to drought than ‘Meski’ [73]. A recent study comparing 32 varieties suggested that ‘Lechin de Sevilla’ and ‘Picholine Marocaine’ were the most drought-tolerant varieties, while ‘Frantoio’ was the most sensitive [74]. ‘Barnea’ and ‘Souri’ trees showed a similar reduction in water potential in our forced drought experiment. In summer drought experiments, the water potential of ‘Barnea’ trees reached lower levels than the ‘Souri’ trees [75]. 

The effect of fruit load on flowering in the ‘no fruit’ sector differed between ‘Manzanillo’ and ‘Coratina’. Since a similar difference was detected in *n* values, it seems that these cultivars differ in perception of fruit load carried on other sectors of the tree. Perhaps the communication between sectors in ‘Coratina’ is stronger so that the different metabolic and hormonal effects of fruit load in one sector also occur in a neighboring sector. This experiment should be repeated in additional trees, using different-sized sectors to clarify such differences between the two cultivars. 

While expression of *OeFT2* in leaves increased during winter in all cultivars, the peak relative expression of the gene was different between cultivars. For example, while *i* values in ‘Coratina’ were higher than ‘Barnea’, *OeFT2* levels in ‘Barnea’ were higher compared to ‘Coratina’. There could be several explanations for this: (a) Perhaps choosing a different sampling date would have shown higher levels of *OeFT2* gene expression in ‘Coratina’. (b) Perhaps the expression of another FT-encoding gene, OeFT1 [10], is higher in ‘Coratina’. (c) Perhaps the level of the *Actin7* housekeeping gene is higher in ‘Coratina’ compared to ‘Barnea’, causing *OeFT2* relative expression levels to appear lower. (d) While we focused here on the positive regulator of flowering, FT, olive meristems contain inhibitors of flowering that can compete with FT, such as the TFL1 family of proteins [76,77,78]. It might be the case that there are more FT-like proteins reaching meristems in ‘Barnea’ compared to ‘Coratina’, yet there may also be higher levels of TFL1-like proteins negating them in ‘Barnea’ compared to ‘Coratina’. Together, this might cause a loss of correlation between levels of *OeFT2* in leaves and *i* values. 

We collected *i* data on trees from different cultivars, with no fruit load memory, in both the FD experiment (Volcani center winter of 2020–2021) and trees exposed to different WTRs in different years and locations. In the FD experiment, ‘Koroneiki’ trees had a significantly higher *i* value compared to ‘Picholine’. In the different WTR experiments, the ‘Askal’ cultivar had significantly higher *i* values compared to ‘Barnea’ after several winters. Interestingly, ‘Askal’ is the progeny of a cross between ‘Barnea’ and ‘Kalamata’, and as such, both cultivars share at least one allele for each locus. Identifying cultivars capable of reaching high-quality oil and high consistent production, with an ability to flower under lower CUs, would likely help Mediterranean olive growers remain competitive under future warmer winters.

## 4. Materials and Methods

### 4.1. Fruit Load Experiments

An initial fruit load experiment was performed on four potted fruit-bearing ‘Barnea’ trees. Potted trees were transferred to a controlled environment of 16/10 °C (day/night) on 27 July 2020 until 11 October 2020, corresponding to the summer season in Israel. Branches with or without fruit (ten each) were marked at the beginning of the experiment. The number of buds formed (*n*) and the percentage of buds that formed a visible inflorescence (*i*) in these branches was evaluated in November 2020. 

The following fruit load experiment (Appendix A) was conducted in the winter of 2021/2022 on ‘Manzanillo’ and ‘Coratina’ cultivars grown outdoors at the Matitiyahu research station in northern Israel (33°03′55.7″ N, 35°27′11.9″ E). Six trees of each cultivar were chosen, and on three of the trees, all fruitlets were removed before 1 June. On the three remaining trees, we identified three major branches emerging from the trunk and defined each branch and the branchlets originating from it as a sector. In the sector designated as having a heavy fruit load, no fruits were removed, and branches marked for the flowering survey had at least four fruits on them. In the medium fruit load sector, up to two fruits were left on each branch with branches marked having exactly two fruits on them. The ‘no fruit’ sector had all of its fruitlets removed before 1 June. 

### 4.2. Winter Forced Drought (FD) Experiment

For the winter drought experiment, mature trees of the following five cultivars were used: ‘Barnea’, ‘Coratina’, ‘Koroneiki’, ‘Picholine’, and ‘Souri’, at the Agricultural Research Organization, Volcani Center in Bet-Dagan Israel (31°59′26.1″ N, 34°49′10.2″ E). A few years before the experiment, the trees were aggressively pruned, such that during the experimental period, they had a significant number of year-old branches with flowering potential and were approximately 150 cm tall. Before the experiment started, they were replanted in large 60-liter pots. During the 2020/2021 winter inductive period (December 2020 through February 2021) and before reproductive bud break, ~20 branches with flowering potential were randomly selected and marked per tree to evaluate the flowering in spring.

For each cultivar, five plants were randomly chosen to be in the control, well-watered group, and the remaining five received the FD treatment. The trees in the control treatment received irrigation via a drip irrigation system (4 drippers with a rate of 2 liters per hour, 2 or 3 times a day for 10 min, altogether 2.66 to 4 liters per day per tree) in addition to natural precipitation. FD-treated plants were kept dry from 24 November 2020 until 12 January 2021 with a mid-winter three-day pause in drought treatment starting 24 December 2020, in which irrigation was provided to avoid tree mortality. The pots of FD-treated trees were covered with a waterproof cloth that did not allow natural precipitation to reach the plants’ soil. Furthermore, they were raised above ground level on cinderblocks such that water could not enter the pots from below and the plants’ roots could not reach the soil underneath the pots (Appendix A).

Specific FD-treated trees (Appendix A) in which the stem water potential dropped lower than −5.0 MPa ended the drought period one week earlier than the rest of the plants in this treatment, with three of these plants receiving an additional temporary dose of one dripper (at two liters per hour) during certain dates in December due to low water potential (Appendix A). 

### 4.3. Measuring Flowering Response of Different Cultivars to Different Winters

Post-winter *I* values in five (‘Askal’, ‘Barnea’, ‘Coratina’, ‘Manzanillo’, and ‘Nabali’) cultivars were recorded starting in the spring of 2015 up to, and including, the spring of 2021. Initially, similar-aged trees were grown in large 25-liter pots. In the 2014–2015 winter, only the data of the ‘Barnea’ trees were recorded in Rehovot, at the Faculty of Agriculture (31°54′15.6″ N, 34°48′04.5″ E). In the winter of 2015–2016, all cultivar data were recorded in Rehovot. Additional trees of similar age were placed at two additional sites: the Gilat ARO station (31°20′01.9″ N, 34°39′52.8″ E) and the Hanania farm (32°56′06.5″ N, 35°25′09.3″ E), and flowering data were collected from the three sites after the winter of 2016–2017. After that, data were collected for the Rehovot winter of 2017–2018. In the fall of 2018, the potted trees were planted in two locations: Rehovot and the Matityahu farm (33°03′55.7″ N, 35°27′11.9″ E), with data in Matityahu collected starting the winter of 2019–2020. In the winter of 2019–2020, ‘Barnea’ trees in Rehovot were not measured due to their poor health. Thus, *i* values for each cultivar were collected in spring after 9 to 11 winters (Appendix A).

The number of biological repeats (trees) for a specific combination of year/location/cultivar varied between three and six. To avoid the inhibitory effect of previous fruit load on *i*, all fruitlets formed on these trees were manually removed before the end of Spring (1 June) of every year. For each tree, 20 appropriate branches were chosen and marked every winter, and *i* was recorded after anthesis in mid-April (see Section 4.5). For all locations and years, hourly temperature data were collected from 15 October until 15 March. Hourly temperatures were calculated by averaging reads collected every 10 min. Temperature data were either measured directly in the field or downloaded from AgriMeteo, a web-based repository of climate data collected from many different weather stations, including ones adjacent to the fields used in these experiments. Based on previous studies [10] we assumed that the period from evocation (the end of flower induction) until the noticeable elongation of inflorescences takes at least two weeks. For each year/location, we recorded when the first signs of inflorescence elongation occurred. In the colder regions of Hanania and Matityahu, this transition occurred ~19 days later than in the warmer regions of Gilat and Rehovot. Since we wanted to provide temperature data until evocation, and not after this transition had begun, we adjusted the end of winter accordingly: February 15th for Gilat and Rehovot and March 6th for Hanania and Matityahu. 

### 4.4. Fruit Removal in Spring 

Each spring, the year before a flowering survey and about a month after the fruit set, all fruitlets were removed (except for those participating in the “Varying fruit load” experiment) from every branch on every tree before 1 June. This practice was carefully carried out to ensure that almost every single fruitlet was removed from the experimental plants to be surveyed for flowering the following spring.

### 4.5. Flowering Survey

During the cold exposure period, after the start of natural winter yet before the spring reproductive bud burst had started, branches with flowering potential were chosen at random. Depending on the experiment, 5–20 branches were chosen per plant. Ribbons were tied to mark the previous season’s growth, with the region expected to flower being above, or apically, to the ribbon. After flowering, *n* and *i* values of intact buds were recorded. Non-intact buds were those affected by a pest of some sort or whose placement was in the axil of a pest-ridden leaf or who were not in the axil of a leaf at all. *i* values were calculated per branch, with the *i* value for a specific tree being the average *i* value of all recorded branches on that tree. A tree, in most cases (not including the fruit load experiment), was considered a biological repeat, and the *i* value of each tree within a treatment was used to calculate the averages and variation within that treatment. *i* varied among winters, with the most extreme differences (0% to 87%) occurring in the same year and cultivar at two different locations.

### 4.6. Quantifying Stem Water Potential 

The water status of trees was measured using the pressure bomb method, previously used in olives [68,79]. For this experiment, stem water potential (the water potential of the trunk of the olive tree as opposed to a smaller branch) was quantified using the Rimad 3000 pressure bomb on nitrogen gas. This was performed by first covering one or two selected branches on each of the trees in the experiment (each tree was designated as a biological repeat) ≥ one hour before measuring the water potential. Branches were covered with special plastic bags lined with tin foil to stop photosynthesis and transpiration such that the water potential of the covered branch would equalize with that of the stem (trunk) of the plant. Water potential was first quantified two weeks after the start of the FD treatment and then weekly for the following five weeks for a total of six measurements during the FD period. Measurements were made at midday (12:00–14:00) to reduce the effect of natural variation in water potential during the diurnal cycle. 

### 4.7. Leaf Sampling for RNA Extraction, cDNA Preparation, and Relative Gene Expression Analysis

Leaf samples were taken from each tree on three sampling dates throughout the FD experiment (8 December 2020, 10 January 2021, and 25 January 2021). The sampling time was approximately five hours after sunrise. Samples taken were from three healthy branches, with the two opposite leaves at the first healthy basal bud position of a branch with flowering potential being chosen for sampling. Thus, a total six leaves were sampled and combined to represent each tree. Leaves were immediately placed in liquid nitrogen after sampling and then stored in a −80 °C freezer until further processing. 

Preparation of samples for RNA extraction was performed by grinding the plant material in a mortar and pestle while ensuring that the tissue was kept frozen at all times using liquid nitrogen. Total RNA extraction was then performed followed by mRNA isolation. Thereafter, cDNA was synthesized, and qPCR was used to quantify relative gene expression levels. All protocols, from the processing of samples through the qPCR analysis, were conducted as previously described [10,11]. Primers for real-time PCR reactions for the *OeFT2* (OE6A103537) gene as well as the *OeACT7* (OE6A117728) housekeeping gene, as presented in Appendix A, are the same as those previously used [10,11].

### 4.8. Cold Unit Calculation

Parameters used for calculating Dynamic model Chilling Portions [44] were the following: E_0_ = 4153, E_1_ = 12,888.8, A_0_ = 139,500, A_1_ = 2.57 × 10^18^, tetmlt = 277, and slp = 1.6.

### 4.9. Statistical Analysis

Differences between treatments were determined by Student’s *t*-test. Statistical significance was determined at *p* ≤ 0.05. Effect Tests reported in Appendix A were conducted using JMP Pro 14 software (SAS Institute, Cary, NC, USA). The effect test for a given effect tests the null hypothesis that all parameters associated with that effect are zero. 

## 5. Conclusions

Olive trees typically exhibit highly variable flowering levels, which affects their yield. Understanding how different WTRs affect the level of *i* in different olive cultivars is essential for predicting the sustainability of olive oil production under warmer winters. In this work, we quantified the separate effects of the previous year fruit load, winter drought, and WTR on *i*. We found that while drought has no effect, both previous fruit load and winter temperature are major determinants of *i*. Studying the effect of WTR on *i* using trees that carried different levels of fruit load is not informative. To focus on maximum *i* levels reached by different genotypes under different WTR regimes, one would need to study trees that had no fruit load in the previous year. Here, we attempted three different models that sum up the warm and cold hours with each temperature being weighted in its effect. The optimal temperatures for olive flower induction have become clearer; yet the underlying mechanism by which warm temperatures during winter disrupt flower induction remains unknown. Unsurprisingly, these models show poor flowering level prediction ability for winters yielding low to intermediate flowering levels, during which warm periods play a major role. This is because the models we used are overlooking the exact distribution of temperatures and specifically the length of continuous warm periods. In conclusion, we propose that further studies clarifying the biological effect of short versus long warm periods are needed. Additionally, a different modeling approach, which also accounts for the length of warm periods, should be developed and tested on the raw data provided here for the different cultivars. 

## Figures and Tables

**Figure 1 plants-12-01714-f001:**
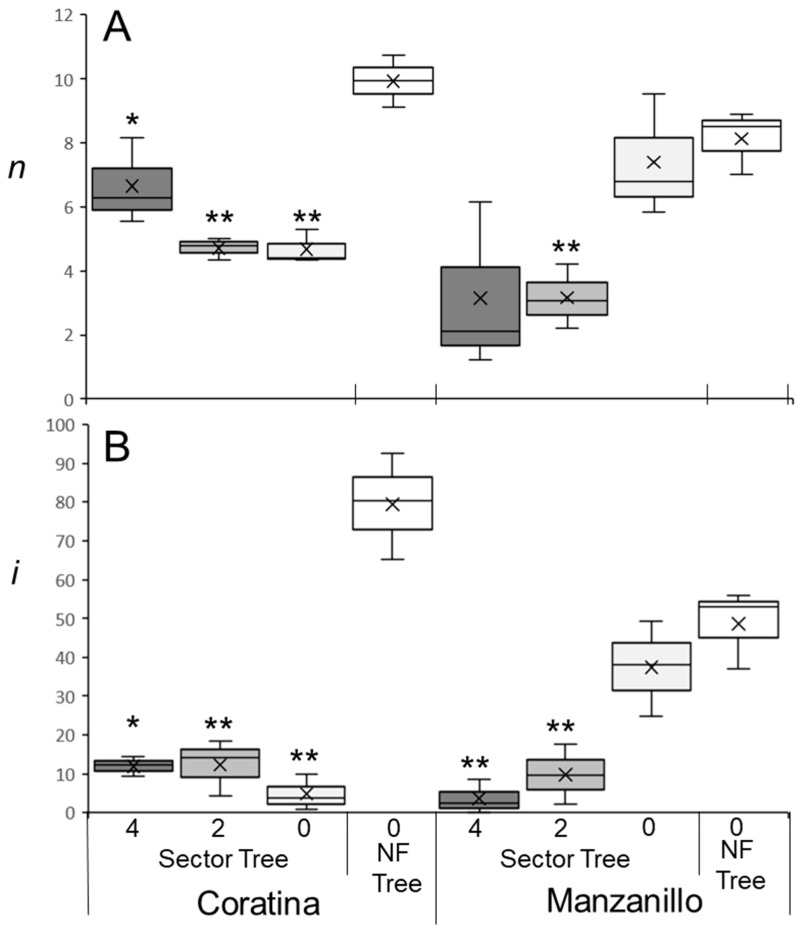
Number of buds per branch (*n*) and the percentage of buds forming inflorescences (*i*) under different levels of fruit load. Experiments were performed on ‘Manzanillo’ and ‘Coratina’ trees at the Matityahu station during the 2021–2022 winter. Fruitlets were removed from trees before 1 June 2021. For each cultivar, all fruitlets were removed (NF Tree) for three trees, while for three additional trees, three similarly sized sectors were defined using three major branches (Sector Tree). In one sector, all fruitlets were removed (0, white). In another sector, fruitlets were thinned so as not to retain more than two fruits on a one-year-old branch (2, light grey). In the third sector, all fruit was retained and the branches chosen contained 4 fruits (4, dark grey). Between 13 and 15 branches in each sector were chosen and marked. In spring, *n* values (**A**) and *i* values (**B**) were recorded for each branch, and the means for each tree were calculated. The medians of the data sets are the lines within the boxes and the means of the data sets are the “X”s found on the plots. Outliers are dots above and/or below the boxes. The figure was created using the inclusive median method. Significant differences between control no-fruit trees and the different sectors of trees from the same cultivar were calculated using the Student’s *t*-test (“*” is *p* ≤ 0.05, “**” is *p* ≤ 0.01). Effect tests of treatment and cultivar and the interactions between them are presented in Appendix A.

**Figure 2 plants-12-01714-f002:**
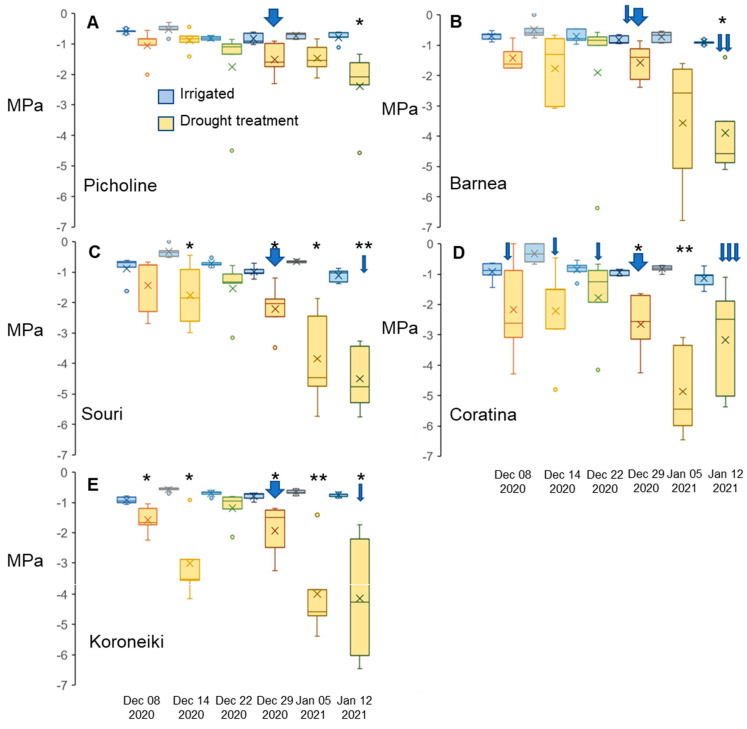
Changes in stem water potential (MPA) under forced drought (FD) treatment. Water potential values are in megapascals (MPa) calculated from one branch per plant, 5 biological repeats (plants) per cultivar per treatment (well-watered and FD) throughout the experimental drought period. We sampled the trees weekly, starting two weeks after the FD treatment began—6 samples in total. A broad blue arrow represents the irrigation of all FD plants before the measuring date. A thin blue arrow represents the irrigation of a single FD plant before the measuring date. The figure was created using the inclusive median method. ‘Picholine’ (**A**), ‘Barnea’ (**B**), ‘Souri’ (**C**), ‘Coratina’ (**D**), and ‘Koroneiki’ (**E**). The medians of the data sets are the lines within the boxes and the means of the data sets are the “X”s found in the plots. Outliers are dots above and/or below the boxes. Using the Student’s *t*-test, dates in which the water potential for FD-treated trees was significantly different compared to control trees on the same date were marked with an asterisk (“*” is *p* ≤ 0.05, “**” is *p* ≤ 0.01). Appendix A contains data for each tree. Appendix A contains the effect tests of the three independent variables in this experiment.

**Figure 3 plants-12-01714-f003:**
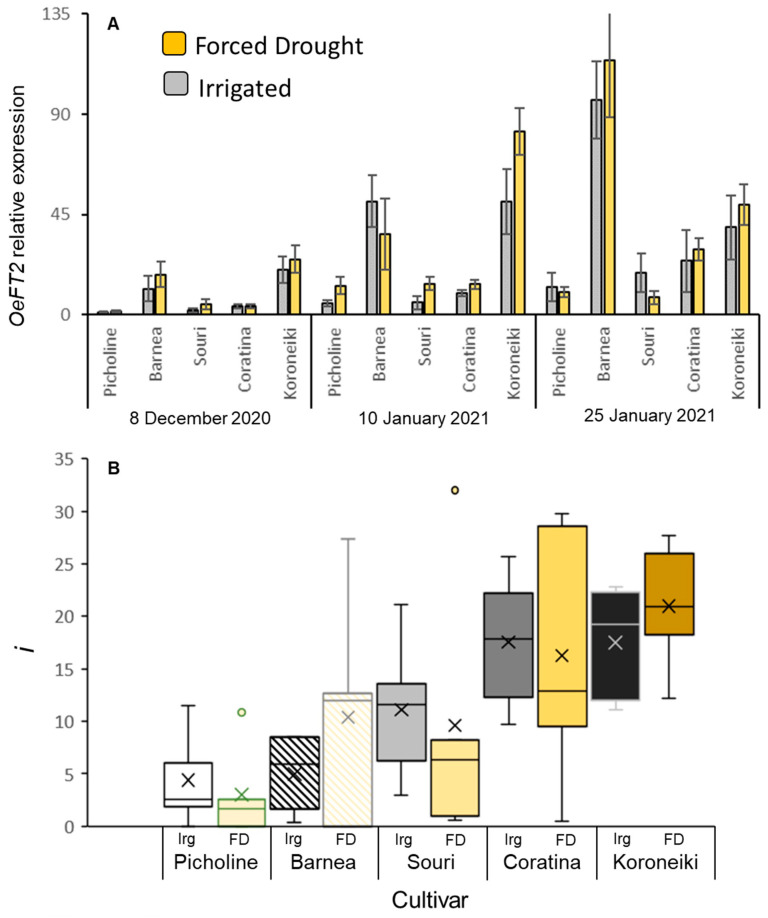
Changes in *OeFT2* gene expression and the percentage of buds forming inflorescences (*i*) under forced winter drought (WD). *OeFT2* relative expression in leaves (**A**) at three time points during winter, in well irrigated (Irg) or FD trees of five cultivars. Means were calculated for 5 biological repeats (trees). The standard error of the mean is presented as bars. *i* values (**B**) were determined for 18–20 branches per tree, and the means for each cultivar and treatment were calculated. Box and whisker plots follow the same format as in Figure 2. Effect tests of treatment date and cultivar and the interactions between them on *OeFT2* expression are presented in Appendix A and on *i* values in Appendix A.

**Figure 4 plants-12-01714-f004:**
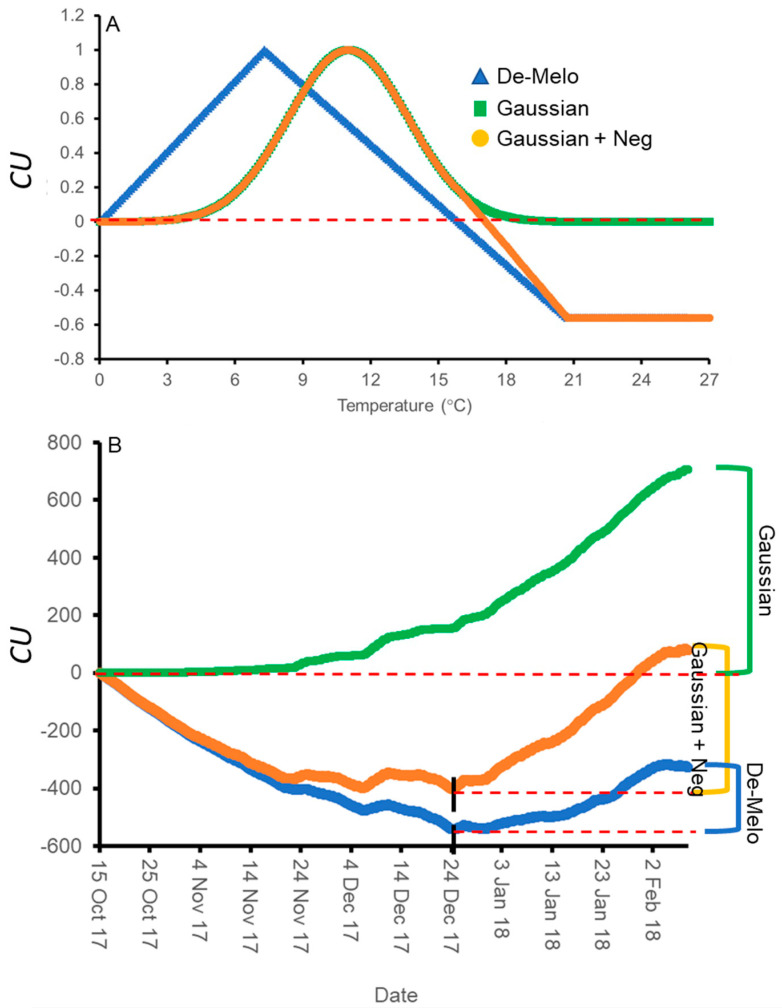
Cold unit (CU) accumulation models under different temperatures and during the winter of Rehovot 2017–2018. Positive and negative CU accumulation as a function of temperatures (**A**) in the three models tested, the DMA model ([36], blue triangles), and the two models developed here: the ‘Gaussian’ (green squares) and ‘Gaussian + Neg’ (orange circles) models. (**B**) Comparing CU accumulation of these three models during the winter of 2017–2018 in Rehovot using recorded hourly temperatures. CU accumulation at the end of winter is defined as the final CU value (in the ‘Gaussian’ model) or the difference between the final CU value at the end of winter and the minimal level of CUs reached during the winter (in the DMA and ‘Gaussian+Neg’ models).

**Figure 5 plants-12-01714-f005:**
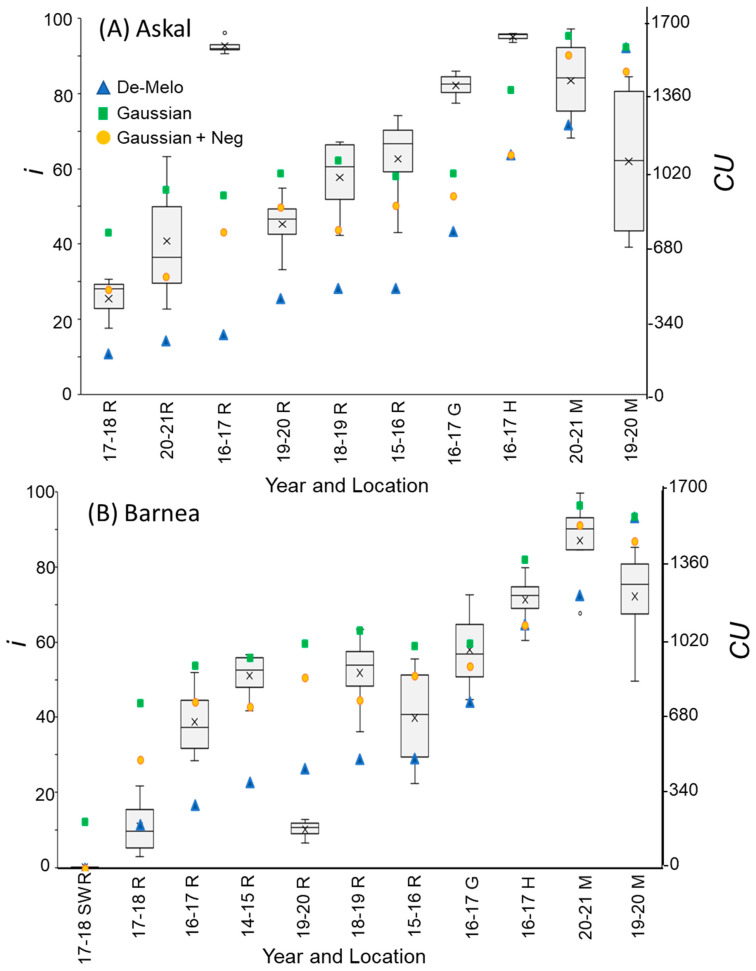
Percentage of buds forming inflorescences (*i*) compared to chill units (CU) accumulation, calculated using 3 different models, under different winter temperature regimes (WTRs) for cultivars ‘Askal’ (**A**) and ‘Barnea’ (**B**). For *i*, box and whisker charts are as described in Figure 1. For each WTR, CU accumulation up to putative evocation was calculated using the DMA model [36] (blue triangles), the ‘Gaussian’ (green squares), and ‘Gaussian +Neg’ (orange circles) models.

**Figure 6 plants-12-01714-f006:**
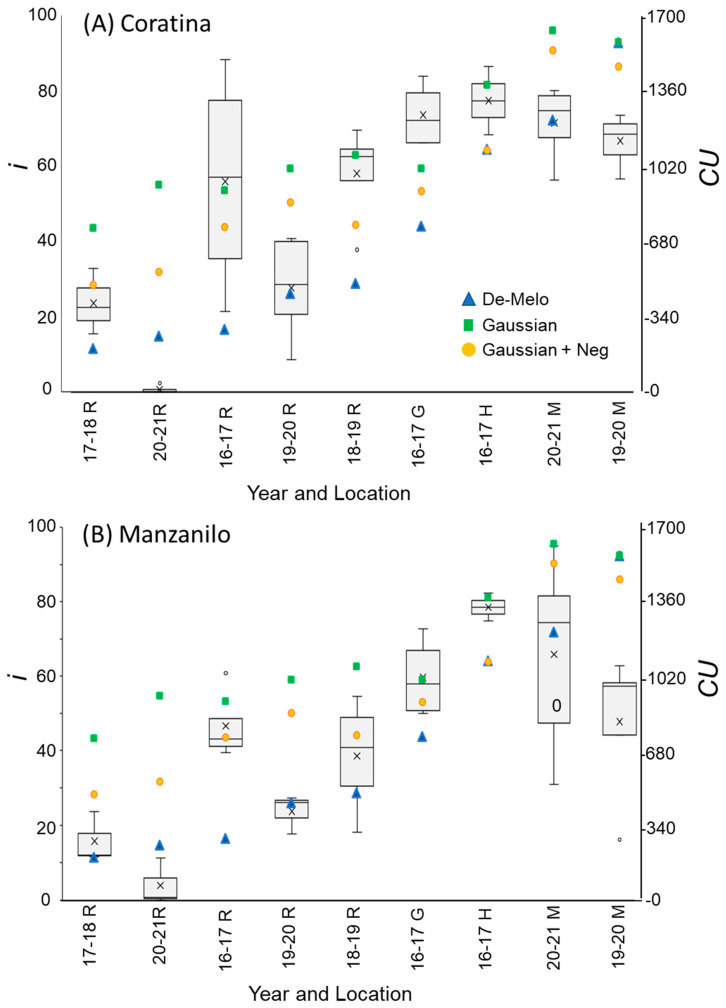
Percentage of buds forming inflorescences (*i*) and chill units (CU) under different winter temperature regimes (WTRs) for cultivars ‘Coratina’ (**A**) and Manzanilo’ (**B**). For *i*, box and whisker charts are as described in Figure 1. For each WTR, CU accumulation up to putative evocation was calculated using the DMA model [36] (blue triangles), the ‘Gaussian’ (green squares), and ‘Gaussian + Neg’ (orange circles) models.

**Figure 7 plants-12-01714-f007:**
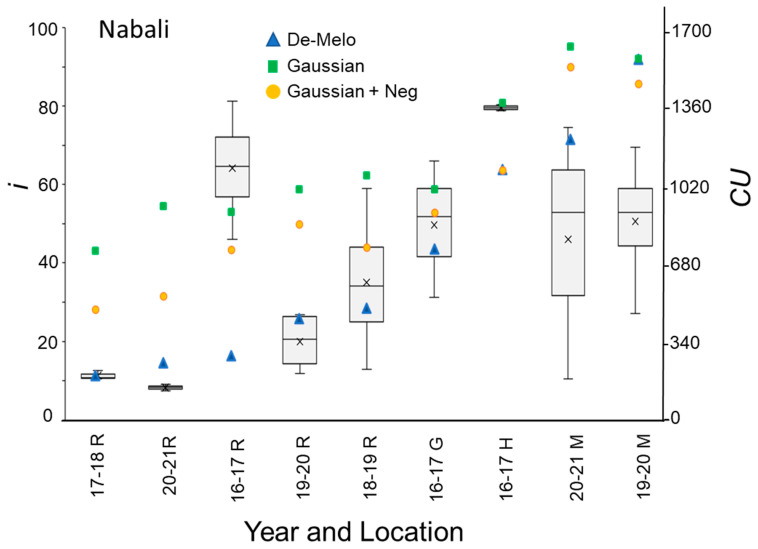
Percentage of buds forming inflorescences (*i*) and chill units (CU) under different winter temperature regimes (WTRs) for the ‘Nabali’ cultivar. For *i*, box and whisker charts are as described in Figure 1. For each WTR, CU accumulation up to putative evocation was calculated using the DMA model [36] (blue triangles), the ‘Gaussian’ (green squares), and ‘Gaussian + Neg’ (orange circles) models.

**Table 1 plants-12-01714-t001:** Calculated CU values accumulated during the 2019–2020 winter and recorded mean *i* values in the following spring.

Location	De Melo-Abreu (CU)	Gaus-Sian (CU)	Gaus-Sian and Neg (CU)	*i* Values of Different Cultivars									
				Askal		Barnea		Coratina		Manzanillo		Nabali	
				AVG	SE	AVG	SE	AVG	SE	AVG	SE	AVG	SE
Rehovot	447	1011	858	45.3	4.5	10.2	1.3	27.6	6.1	23.6	3	19.9	3.8
Matityahu	1579	1581	1473	62	11.6	72.2	5.3	66.7	2.7	47.8	8.5	50.6	8.8
T. test between locations				NS		***		**		NS		*	

Asterisks denote significant differences between locations (* *p* ≤ 0.05, ** *p* ≤ 0.01, *** *p* ≤ 0.001) by Student’s *t*-test. NS is not significant.

**Table 2 plants-12-01714-t002:** Calculated CU values, including Dynamic Model Chill Portions (CP) accumulated during different WTRs and measured mean *i* values for ‘Askal’ and ‘Barnea’.

WTR	De Melo (CU)	Gaus-Sian (CU)	Gaus-Sian and Neg (CU)	Dyn-Amic (CP)	*i* Values of Different Cultivars				
					‘Askal’		‘Barnea’		*t*-Test between Cultivars
					AVG	SE	AVG	SE	
2015–2016 Rehovot	493	999	866	20.3	97.2	2.4	39.9	7.7	NS
2016–2017 Rehovot	284	912	747	13.1	92.7	1.2	38.8	5.2	**
2016–2017 Gilat	749	1011	909	33.8	82.1	1.9	62.5	7.3	NS
2016–2017 Hananya	1097	1388	1095	55.9	95.1	0.8	71.3	4	**
2017–2018 Rehovot	197	743	488	8	25.4	4	11	4.2	NS
2018–2019 Rehovot	491	1071	757	18.1	57.7	5.8	51.8	5.8	NS
2019–2020 Rehovot	447	1011	858	20.1	45.3	4.5	10.2	1.3	**
2019–2020 Matityahu	1579	1581	1473	72.5	62	11.6	72.2	5.3	NS
2020–2021 Matityahu	1229	1633	1546	55.8	83.4	6.5	87	5.4	NS

Asterisks denote significant differences between cultivars within a location ** *p* ≤ 0.01, by Student’s *t*-test. NS is not significant.

## Data Availability

Data is contained within the article or Appendix A.

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
