# Peer review of "Studying Parameters Affecting Accumulation of Chilling Units Required for Olive Winter Flower Induction"

_plants, 2023, doi:10.3390/plants12081714_

Round 1
Reviewer 1 Report
In the manuscript entitled ¨Unraveling the Impact of different temperatures on the accumulation of chilling units required for olive winter flower induction¨ by Engelen et al., authors showed an enormous amount of results from very complex experiments. The subject is important and of interest. However, the text needs serious improvement concerning English writing. Experimental design is very complex and difficult to understand. Maybe, authors should consider keeping just a part of the experiments to be able to make a smooth story. Some specific comments bellow.
Please, number pages and lines to facilitate the revision process.
Abstract: Reorganize abstract.
Introduction
2nd paragraph: ¨…y between 30-0.3 Tons s, with ~20% of fruit weight later on producing olive oil¨: How many tons? It is not understandable that. Rewrite sentence after the comma because is also not understandable.
2nd paragraph: ¨Assuming an average of 15 flowers per inflorescence, the above tree will likely contain ~ 33,333 inflorescences.¨ Replace ¨above¨ per ¨average¨. I can not understand how the authors arrive to these number of inflorescences per tree. There is a mistake somewhere.
2nd paragraph: ¨Only 0-94% of inflorescences will retain fruit [10], and the average number of fruit on such inflorescences will range between one and two [10].¨ Again it is very complicated to understand the numbers. 0-94% is a very broad range of variation. One or two fruit per inflorescence is also a weird number.
4th paragraph: It contains very important information, but it is confused. Please, revise and rewrite it.
5th paragraph: names of fruits should be written with minuscule letters.
6th paragraph: FT role.
7th paragraph: What is the difference between WTR and chilling requirement? This should be clear in the introduction.
Last paragraph of the introduction is entangled. This contains a resume of the experimental design, but it is not clear. Maybe it is because the text needs English edition.
Material and methods
2.1 What is the i value? Highlight the seasons corresponding to each date. There are several readers from Southern hemisphere.
The experimental design is very complex. It is very difficult to understand just how it is written. Authors used different varieties of olive during different harvest seasons with treatments in different moments of the year. Text should be improved and maybe a scheme would be useful. Authors should also consider if they want to keep all the experiments in the manuscript.
Footnote of Table 3 do not correspond to table´s content.
Results
Results section should contain just a description of the figures and tables showed. There is a lot of discussion and interpretation as it is written. Again, I think that authors should evaluate if they will keep all the results in the manuscript. It is important to construct a wrapped story.
Quality of tables and figures should be improved.
Revise references, there are several mistakes.
Scientific names should be written in italic with genus name with capital letter initial and species minuscule letter.
Author Response
Thank you for your very helpful comments. See attached for detailed response

Author Response
Thank you for your helpful comments. Attached are the changes we made, according to your suggestions.
